# Childhood Pemphigus Vulgaris during COVID-19 Outbreak Successfully Treated with Prednisone and Azathioprine: A Case Report and Literature Review

**DOI:** 10.3390/jcm11226858

**Published:** 2022-11-21

**Authors:** Pamela Vezzoli, Michele Parietti, Andrea Carugno, Marco Di Mercurio, Chiara Benaglia, Martina Zussino, Riccardo Cavalli, Paolo Sena, Emilio Berti

**Affiliations:** 1Dermatology Unit, ASST Papa Giovanni XXIII Hospital, 24127 Bergamo, Italy; 2Dermatology Unit, Department of Medical Sciences, University of Turin, 10124 Turin, Italy; 3Ph.D. Program in Molecular and Translational Medicine (DIMET), University of Milan-Bicocca, 20126 Milan, Italy; 4Dermatology Unit, Foundation IRCCS Ca’ Granda Ospedale Maggiore Policlinico, Università degli Studi di Milano, 20122 Milan, Italy; 5Department of Pathophysiology and Transplantation, Università degli Studi di Milano, 20122 Milan, Italy; 6Pediatric Dermatology, Foundation IRCCS Ca’ Granda, Ospedale Maggiore Policlinico, Università degli Studi di Milano, 20122 Milan, Italy

**Keywords:** childhood pemphigus, autoimmune blistering disease, immunosuppressive therapy

## Abstract

Introduction: pemphigus vulgaris is a rare autoimmune blistering disease that involves the skin and mucous membranes and rarely occurs in pediatric age. Methods: we present a case of childhood pemphigus in a 9-year-old patient from Burkina Faso, which initially manifested with erosive lesions symmetrically distributed in the oral cavity. After a few months, we also observed hyperchromic lesions of the back. Histopathological examination of skin samples showed intraepidermal acantholysis, while direct immunofluorescence showed deposits of complement (C3) and immunoglobulins G (IgG) in the epidermidis; an ELISA test highlighted the presence of circulating autoantibodies against desmoglein 3. Results: the follow-up of this patient was made difficult by the advent of the COVID-19 outbreak. However, after about one year of combined therapy with systemic steroids and azathioprine the patient reached clinical remission.

## 1. Introduction

Pemphigus Vulgaris (PV) is an autoimmune blistering disease that involves skin and mucous membranes. It is a rare disease in the general population (1:1,000,000) and even rarer in pediatric age. Childhood Pemphigus Vulgaris (CPV) is a pediatric variant of PV, which affects children below 12 years [1]. Clinical presentation includes flaccid blisters that produce painful erosions after rupture. The majority of patients with CPV have lesions in both mucosa and skin, with the appearance of oral lesions being the first manifestation of the disorder. Mucous membranes of the anus, conjunctiva, and genital areas are frequently involved [2]. We present a case of CPV in a young male that came to our attention for the onset of painful oral erosions just before the COVID-19 outbreak, which hit the city of Bergamo in March 2020.

## 2. Case Report

A 9-year-old male from Burkina Faso was admitted to our Dermatology Unit for the onset of extensive erosions localized in the oral cavity. The manifestation began twenty days earlier, accompanied by low-grade fever, malaise and bilateral conjunctivitis. Antiviral therapy prescribed by the pediatrician did not lead to an improvement in symptoms. At clinical examination, the child showed erosions on the upper and lower lips, on the surface of the tongue, on the hard palate, on the soft palate and symmetrically on the cheek mucous membranes (Figure 1a–c). Feeding difficulties, diarrhea or constipation were denied. Laboratory tests, mycoplasma and herpes serologies were normal. There was no history of drug intake in the previous fifteen days. In the clinical hypothesis of an autoimmune blistering disease, we performed mucosal biopsy at the left cheek oral mucosa. Histopathological examination showed acantholysis and intact basal layer, which are typical features of PV (Figure 2a,b). Direct immunofluorescence (DIF) gave a negative result on this sample, but an enzyme-linked immunosorbent assay (ELISA) test showed the presence of high titre circulating autoantibodies against desmoglein 3 and negativity of autoantibodies against desmoglein 1; desmocollins were not tested. Even in the absence of a positive DIF we made a diagnosis of CPV. The boy weighed 35 kg. We started systemic steroid therapy with betametasone 4 mg daily decreasing in four weeks, until the next follow-up visit. Unfortunately, we were unable to visit the patient for a long time, due to the COVID-19 outbreak that hit the city of Bergamo in the spring of 2020.

Six months later, when the lockdown was lifted, the patient returned to our unit with a worsening of the mucosal manifestations associated with fetor, hoarseness and moderate dysphagia. Clinical examination showed two hyperpigmented patches on his left shoulder, not present at the previous visit (Figure 1d). We performed another incisional biopsy at this site on suspicion that the lesions were manifestations of the same pathology; on this sample we found at direct immunofluorescence a positivity for C3 and IgG deposits with a fishnet pattern (Figure 2c,d). These findings supported the diagnosis of CPV. Steroid therapy alone appeared to give only minimal improvement in the clinical status of the patient. Indeed, after one month of a new course of steroid therapy, gum and tongue erosions were still present. The patient did not have COVID-19; neither did he have the suspected symptoms of COVID-19. In any case, we were doubtful about adding an immunosuppressant to the current therapy; in fact, we still had little data regarding the possible effects of a concomitant COVID-19 infection. For the same reason, we did not consider the option of starting a course of rituximab therapy. Considering the progressive deterioration of the patient, we decided to add azathioprine 75 mg daily (approximately 2 mg/kg) to oral steroid therapy (prednisone 1 mg/kg daily). We began to see improvement in the mucosal lesions, and after one month we set a very slow tapering of the steroid. The patient continued the treatment with oral steroids and azathioprine for more than one year and finally he gained clinical remission (Figure 3). To date, after 18 months of follow-up, the child has normal growth, normal weight gain, no Cushing’s syndrome and maintains clinical remission.

## 3. Discussion

The appearance of pemphigus vulgaris before the age of 18 accounts for 1.4 to 3.7% of all cases of PV. It affects women more than men. Pemphigus vulgaris in pediatric age can be defined as Juvenile PV (JPV) in patients older than 12 years old, and Childhood PV (CPV) in children under 12 years old [3]. Childhood pemphigus vulgaris is much rarer than JPV, and few cases have been described in the literature (approximately 50 cases, with onset age range 1.5–12 years). Childhood pemphigus vulgaris in the initial stages usually occurs with an important involvement of the oral mucosa with multiple painful vesicles and eroded lesions, as presented in our case. After some weeks or months, the manifestation can extend to other mucous membranes and skin. A delay in diagnosing CPV often occurs, due to the rarity of PV in this age group and a similar clinical appearance with other bullous and ulcerative diseases that affect the oral cavity. The wide variety of prevalent blistering diseases and the lack of familiarity with this entity are the main causes of diagnostic delay.

The clinical differential diagnosis of CPV affecting mucocutaneous tissues includes recurrent aphthous stomatitis, acute herpetic gingivostomatitis, bullous epidermolysis, linear IgA disease, paraneoplastic pemphigus, erythema multiforme, cicatricial pemphigoid and erosive lichen planus [4]. In our case, the manifestation was initially mistaken for herpetic infection, and this led to a delay, albeit minimal, in the diagnosis.

Although the exact etiology remains obscure, a wide range of antigenic factors, including drugs, herpetic and bacterial infections and malignancy have been suggested as triggering factors [5].

There are few differences between adult and childhood PV, as they have similar etiopathogenesis. The main difference is the higher incidence of genital and conjunctival involvement in the pediatric variant. Also, there is no difference between PV and CPV concerning diagnostic tests [6]. The presence of autoantibodies against desmogleins 1 and 3 at indirect immunofluorescence is quite common. Histological examination remains mandatory in cases of difficult clinical interpretation. Prognosis in children seems to be better than in adults, but it may remain reserved due to the potentially harmful side effects of the steroids and immunosuppressive therapy. Deaths from pemphigus vulgaris in pediatric age are usually caused by massive cutaneous involvement (greater than 70%), sepsis, pneumonia and serum electrolyte alterations [2,7].

The principal therapies used in CPV are immunosuppressants, in a dose adjusted according to age, weight, disease severity and drug side-effects in the young patient. Prednisone is usually the first-choice therapy, and its dosage is adjusted according to the response and gradually tapered in cases of clinical improvement (1 to 2 mg/kg/day). Treatment can be stopped when lesions disappear; however, a maintenance dose may be required in some patients. When high doses of steroids are used for long-term treatment, it is recommended to add adjuvant therapy such as mycophenolate mofetil and azathioprine [8]. Adjuvants have a steroid-sparing effect, allowing to avoid side effects like weight gain, low development, hypertension and diabetes. They may also lead to steroid-free remission. Rituximab, a monoclonal antibody against CD20, has recently been approved as first-line therapy for PV and now widely used [9]. Also in children refractory to conventional therapy or steroid dependent, rituximab is available since 2018 in USA and 2019 in Europe [10]. In addition to these conventional therapies other biologic agents are currently in clinical progress for the treatment of PV: Efgartigimod, a neonatal FcRn receptor inhibitor, is currently in clinical progress for the treatment of PV, it may reduce the levels of circulating autoantibody in the serum [11]. Furthermore, T cells expressing chimeric autoantibody receptors (CAAR-T cells) seem to be one the latest proposed avenues for treating PV [12]; CAAR- T cells may provide an effective and universal strategy for specific targeting of autoreactive B cells in antibody-mediated autoimmune disease PV [12].

Great results were obtained in the treatment of CPV with the association of steroids and azathioprine as reported in Table 1. Our initial concerns in prescribing azathioprine were related to the concomitant outbreak of COVID-19, since immunosuppressive therapy can generally inhibit antiviral immunity and patients undergoing immunomodulatory treatment may be at higher risk of worse outcomes should they develop COVID-19. Indeed, it has been reported that autoimmune bullous disease patients treated with rituximab within the last year may have more severe/prolonged COVID-19 compared to healthy people [13].

Moreover, as specified by the same authors, the initiation of rituximab in autoimmune blistering disease patients must be weighed against the risks of conventional immunomodulatory regimens [13]. Rituximab has been shown to impair humoral responses for 6 months or longer post-administration [14]. This could have represented a problem in view of a possible vaccination. On the other hand, a systematic review of the infection risk, vaccine response and management strategies of pemphigus with immunomodulating therapies including rituximab during the COVID-19 epidemic [15] did not reveal a higher rate of COVID-19 infection in pemphigus patients; furthermore, this paper underlines that, to date, data concerning Rituximab during the pandemic are still insufficient. Finally, although vaccines may trigger or aggravate the pemphigus course, it has not yet been clarified whether COVID-19 vaccines should be held accountable for pemphigus and, given the risk of hospitalization and death associated with COVID-19, vaccination still ought to be advocated in most patients [15]. Indeed, in our experience of patients with bullous disease during the first wave of the COVID-19 pandemic (62 with BP and 31 with PV), we observed a stable disease activity in all the patients and ongoing steroid or immunosuppressive therapy has been stopped for hospitalized patients [16].

In the end, given the progressive clinical worsening of the patient, we opted to use azathioprine rather than rituximab.

## 4. Conclusions

The case presented is relevant for the rarity of this disease in pediatric age. Information on typical clinical elements can be useful to those who are faced with similar scenarios, often difficult to interpret. To these peculiarities is added the difficulty in managing the case during the COVID-19 epidemic. In fact, there was little information regarding the use of immunosuppressive therapies and their potential interaction with this viral infection. The use of combination therapy with steroid and azathioprine allowed us to reduce the infectious risk and achieve prolonged clinical remission. We believe that this case makes an additional contribution to the knowledge of CVP, helping to achieve this difficult diagnosis and leading to its appropriate treatment.

## Figures and Tables

**Figure 1 jcm-11-06858-f001:**
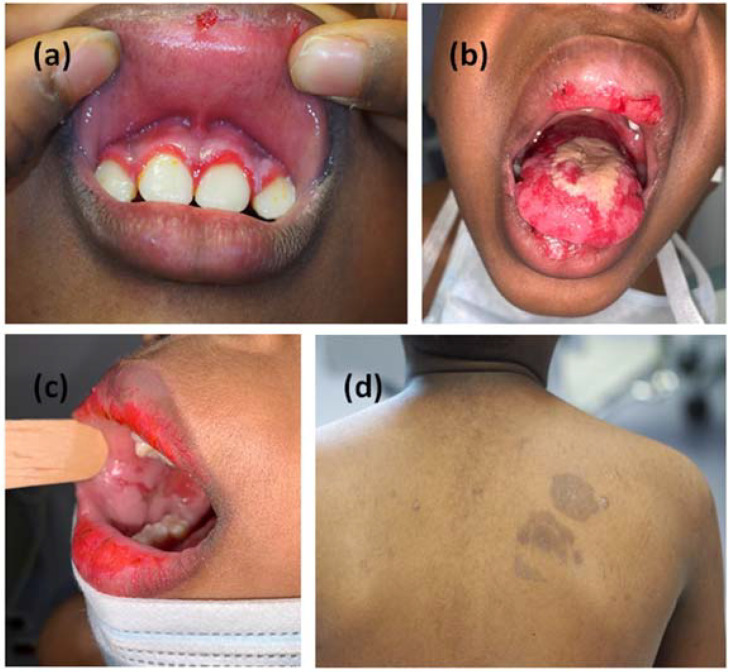
Clinical manifestation with erosive lesions of the oral cavity symmetrically distributed to gum (**a**), tongue (**b**) and mucous membrane of the cheeks (**c**). Hyperpigmented lesions on the skin arose six months later (**d**).

**Figure 2 jcm-11-06858-f002:**
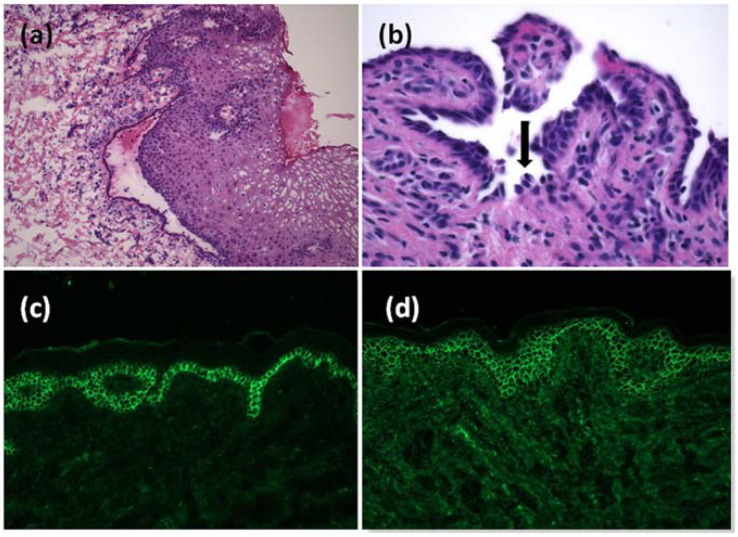
Histopathological examination of the mucosa at small magnification showed a detachment of the epithelium from the chorion (**a**). Acantholysis in histological examination of the skin, with some keratinocytes looking similar to tombstones (black arrow) (**b**). Positivity for C3 (**c**) and IgG (**d**) deposits showing a fishnet pattern at direct immunofluorescence on the skin biopsy.

**Figure 3 jcm-11-06858-f003:**
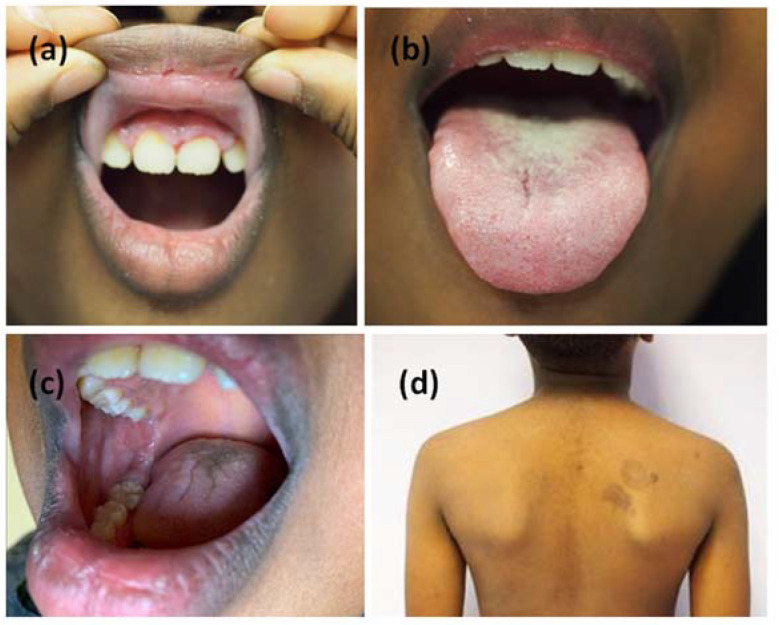
Complete remission of the mucous lesions (**a**–**c**). Pigmented outcomes on the skin (**d**).

**Table 1 jcm-11-06858-t001:** Description of similar cases of CPV treated with steroids and azathioprine reported in the literature.

Authors	Sex	Patient Age at Diagnosis	Clinical Features	Mucosal Involvement	Histopathology/DIF	IIF/ ELISA	Treatment	Follow-Up
Current case	M	9	Erosions on the upper and lower lips, on the surface of the tongue, on the hard palate, on the soft palate and symmetrically on the cheek mucous membranes. Limited lesions on the back	Oral	Acantholysis and intact basal layer. Intercellular IgG and C3	Anti-DSG 3 (174.4 U/mL) *	Prednisone 1 mg/kg daily, Azathioprine 2 mg/kg daily	Clinical improvement in 1 month. Still on prednisone and azathioprine after 18 months
Virtuoso, J. [2]	M	10	Erosive, bleeding painful lesions of oral mucosa, sialorrhoea, weight loss, dry cough	Oral	Intraepidermal acantholysis. Intercellular IgG and C3	Anti-DSG 3	Prednisolone 2 mg/kg daily, Azathioprine 0.5 mg/kg daily	Tapered off drugs after 8 months; no maintenance therapy
Wananukul, S. [3]	F	12	Vescicobollous lesions at trunk, extremities and scalp. Extent of involvement 30%	Oral	Suprabasal separation containing acantholytic cells. Intercellular IgG	1:160	Prednisolone 5 mg/kg daily, Azathioprine 2 mg/kg daily	Tapered off drugs in 7 months; no lesions for 3 years; no maintenance therapy
Wananukul, S. [3]	F	11	Vescicobollous lesions at scalp, face, trunk and extremities. Extent of involvement 15%	Oral and genital	Suprabasal separation containing acantholytic cells; intercellular IgG and C3	1:640	Prednisolone 3 mg/kg daily, Azathioprine 2 mg/kg daily	Tapered off drugs in 12 months, lesion-free for 15 months
Fuertes, I. [4]	F	1.5	Trunk	Oral	Suprabasal separation containing acantholytic cells; intercellular IgG and C3	Not done	Prednisolone 6 mg/kg daily for 1 month, Azathioprine 1–4 mg/kg day for 4 months, Cyclosporin 5–6.5 mg/Kg/day, Dapsone, Prednisone 20–40 mg/Kg/day, Rituximab	Infection
Gupta, N.T. [17]	M	7	Generalized scaling and redness associated with pedal oedema and tightness of both upper and lower extremities. Flaccid blisters on scalp, face and both lower extremities associated with photosensitivity	No	Suprabasal separation containing acantholytic cells. IgG and C3 at the basement membrane zone	Anti-DSG 1 and 3	Prednisolone 2.5 mg/kg daily, Azathioprine 25 mg daily	Complete remission. Still on prednisolone 5 mg daily and azathioprine 50 mg daily
Bean, S.F. [18]	M	11	Generalized bullous eruption	Erosions in mucosa	Suprabasal separation containing acantholytic cells; intercellular IgG and C3	IgG 1:1280	Prednisone, Azathioprine (dosage not specified)	More than 3 years, multiple relapses
Kanwar, A.J. [19]	M	14	Vesiculobullous lesions at scalp, face, trunk and extremities. Extent of involvement 50%	Oral (anus at 9 months after diagnosis)	Suprabasal separation containing acantholytic cells; intercellular IgG and C3	1:160	Prednisolone 3 mg/kg daily, Azathioprine 2 mg/kg daily	Still on prednisolone 20 mg (alternate days) and azathioprine (1 mg/Kg/day) at 2 years; intralesional corticosteroids at vegetation
Harangi, F. [20]	M	10	Genital and trunk lesions	Oral (anus at 9 months after diagnosis)	Suprabasal separation containing acantholytic cells; intercellular IgG and C3	Not done	Prednisolone 1.5–2 mg/kg daily, Azathioprine 2 mg/kg daily for 4 months	Complete remission for 4 years
Kong, H.H. [21]	F	10	Trunk (20% of body surface)	Oral	Suprabasal separation containing acantholytic cells; intercellular IgG and C3	Not done	Prednisone 1.6–2 mg/kg daily, Azathioprine 2 mg/kg daily for 4 months, Mycophenolate mofetil 1–2 g/day, Intravenous immunoglobulin, Rituximab	Infection
Koturoglu, G. [22]	F	11	Trunk, extremities	Oral	Suprabasal separation containing acantholytic cells; intercellular IgG and C3	Not done	Prednisolone 1.5–2 mg/kg daily, Azathioprine 2 mg/kg daily for 6 months	Partial remission on maintenance therapy

* Normal value: <7 U/mL. Abbreviations: DIF: Direct Immunofluorescence; IIF: Indirect Immunofluorescence; ELISA: Enzyme Linked Immunosorbent Assay; IgG: Immunoglobulin G; C3: Complement 3; DSG: desmoglein.

## Data Availability

Data available upon request to corresponding author.

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
