# Peer review of "Childhood Pemphigus Vulgaris during COVID-19 Outbreak Successfully Treated with Prednisone and Azathioprine: A Case Report and Literature Review"

_jcm, 2022, doi:10.3390/jcm11226858_

Round 1
Reviewer 1 Report
The paper by Vezzoli et al. describes the diagnosis and treatment of childhood pemphigus vulgaris (CPV) in a 9 year old boy in Italy during the first year of the COVID-19 pandemic.
The boy was successfully treated with a combination therapy of steroids and the immunosuppressant azathioprine.
Specific comments:
· The patient was described to have autoantibodies against desmoglein 3. Does he also have other autoantibodies typical for PV (desmoglein 1 and desmocollins)?
· How long was the initial treatment with betametsone alone?
· The authors justify their decisions with the ongoing COVID-19 pandemic and the putative risk of a severe outcome if the patient would be infected with COVID-19:
o Did the patient have COVID-19, and if yes what were the symptoms?
o Has the patient been vaccinated against COVID-19 and what was his antibody titer?
· The authors discuss the current treatment options for PV, but they should also include newer treatment options in their discussion such as the targeting of FcRn with Efgartigimod.
Minor comments:
· The authors used different ways of writing “covid-19”. This should be done in a consistent way.
· The patient was described as “black”. I am not sure about the correct nomenclature of skin tones.
Author Response
Thank you for your appreciation and suggestion. Here are our answers:
The paper by Vezzoli et al. describes the diagnosis and treatment of childhood pemphigus vulgaris (CPV) in a 9 year old boy in Italy during the first year of the COVID-19 pandemic. The boy was successfully treated with a combination therapy of steroids and the immunosuppressant azathioprine.
Specific comments:
· The patient was described to have autoantibodies against desmoglein 3. Does he also have other autoantibodies typical for PV (desmoglein 1 and desmocollins)?
No, he doesn’t have other autoantibodies typical for PV. Desmocollins were not tested.
· How long was the initial treatment with betametasone alone?
The initial treatment with betamethasone was long four weeks.
· The authors justify their decisions with the ongoing COVID-19 pandemic and the putative risk of a severe outcome if the patient would be infected with COVID-19:
· Did the patient have COVID-19, and if yes what were the symptoms?
No, the patient didn’t have COVID-19 neither he had symptoms suspected for COVID-19
· Has the patient been vaccinated against COVID-19 and what was his antibody titer?
Yes, the patient was vaccinated against COVID-19 one year after the disease, when the pediatric vaccine was available but we did not measure antibody titer.
· The authors discuss the current treatment options for PV, but they should also include newer treatment options in their discussion such as the targeting of FcRn with Efgartigimod.
We have include the targeting of FcRn such as Efgartigimod as a possible treatment options for PV in the discussion as suggested by You.
Minor comments:
· The authors used different ways of writing “covid-19”. This should be done in a consistent way.
We used “covid-19” in a consistent way.
· The patient was described as “black”. I am not sure about the correct nomenclature of skin tones.
We deleted “black” in our manuscript.
Reviewer 2 Report
The authors present a rare Pemphigus Vulgaris (PV) case in a 9-year-old black boy diagnosed and treated during the COVID-19 outbreak. The authors emphasize the difficulties and delay in treatment during follow-up mainly due to the fear of contracting infections.
This case may be suitable for publication in JCM because it accurately describes the clinical, histological and immunofluorescent characteristics of PV in childhood and reports the positive experience of prednisone and azathioprine treatment in this disease.
The report's title envisages the review of similar cases in the literature, which could be more accurate and readable. The authors could summarize the most salient clinicopathological features and the relative treatment in a Table.
Furthermore, considering the ongoing epidemic, the work may become more interesting if the authors even briefly report any differences concerning PV in patients infected or vaccinated for COVID-19, if described in the literature.
Author Response
Thank you for your appreciation and suggestion. Here are our answers:
The authors present a rare Pemphigus Vulgaris (PV) case in a 9-year-old black boy diagnosed and treated during the COVID-19 outbreak. The authors emphasize the difficulties and delay in treatment during follow-up mainly due to the fear of contracting infections.
This case may be suitable for publication in JCM because it accurately describes the clinical, histological and immunofluorescent characteristics of PV in childhood and reports the positive experience of prednisone and azathioprine treatment in this disease.
The report's title envisages the review of similar cases in the literature, which could be more accurate and readable. The authors could summarize the most salient clinicopathological features and the relative treatment in a Table.
We summarized the cases most similar to our patient in Table 1, focusing on the clinicopathological features and treatment with oral steroids and azathioprine.
Furthermore, considering the ongoing epidemic, the work may become more interesting if the authors even briefly report any differences concerning PV in patients infected or vaccinated for COVID-19, if described in the literature.
We briefly report the risk of infection, the role of Covid-19 vaccination and the management strategies of pemphigus during the Covid-19 epidemic as reported by the literature, also describing our experience in autoimmune bullous disease during the first wave of Covid-19 pandemic.
Round 2
Reviewer 1 Report
All my questions have been answered. Thank you.
Author Response
Thank you